# Prediction of Intracranial Infection in Patients under External Ventricular Drainage and Neurological Intensive Care: A Multicenter Retrospective Cohort Study

**DOI:** 10.3390/jcm11143973

**Published:** 2022-07-08

**Authors:** Pengfei Fu, Yi Zhang, Jun Zhang, Jin Hu, Yirui Sun

**Affiliations:** 1Department of Neurosurgery, Huashan Hospital, Fudan University, Shanghai 200040, China; wscxzr@163.com (P.F.); doctorzhang_0221@163.com (J.Z.); 2Engineering Research Center of Traditional Chinese Medicine Intelligent Rehabilitation Ministry of Education, Shanghai University of Traditional Chinese Medicine, Shanghai 201203, China; zhangyi@shutcm.edu.cn; 3Shanghai Clinical Medical Center of Neurosurgery, Shanghai 200040, China; 4National Center for Neurological Disorders, Shanghai 200031, China

**Keywords:** external ventricular drainage, intracranial infection, lasso regression, logistic regression, nomogram, machine learning

## Abstract

*Objective*: To generate an optimal prediction model along with identifying major contributors to intracranial infection among patients under external ventricular drainage and neurological intensive care. *Methods*: A retrospective cohort study was conducted among patients admitted into neurointensive care units between 1 January 2015 and 31 December 2020 who underwent external ventricular drainage due to traumatic brain injury, hydrocephalus, and nonaneurysmal spontaneous intracranial hemorrhage. Multivariate logistic regression in combination with the least absolute shrinkage and selection operator regression was applied to derive prediction models and optimize variable selections. Other machine-learning algorithms, including the support vector machine and K-nearest neighbor, were also applied to derive alternative prediction models. Five-fold cross-validation was used to train and validate each model. Model performance was assessed by calibration plots, receiver operating characteristic curves, and decision curves. A nomogram analysis was developed to explicate the weights of selected features for the optimal model. *Results*: Multivariate logistic regression showed the best performance among the three tested models with an area under curve of 0.846 ± 0.006. Six variables, including hemoglobin, albumin, length of operation time, American Society of Anesthesiologists grades, presence of traumatic subarachnoid hemorrhage, and a history of diabetes, were selected from 37 variable candidates as the top-weighted prediction features. The decision curve analysis showed that the nomogram could be applied clinically when the risk threshold is between 20% and 100%. *Conclusions*: The occurrence of external ventricular-drainage-associated intracranial infections could be predicted using optimal models and feature-selection approaches, which would be helpful for the prevention and treatment of this complication in neurointensive care units.

## 1. Introduction

External ventricular drainage (EVD) is one of the most common lifesaving procedures performed in neurological intensive care units [1]. Various types of brain injuries, including traumatic brain injury, intracranial hemorrhage, subarachnoid hemorrhage, hydrocephalus, and meningitis, may benefit from the insertion of EVD. For patients in critical conditions, EVD allows for continuous or intermittent cerebrospinal fluid (CSF) drainage, by which means therapeutic purposes may be achieved such as intracranial pressure (ICP) monitoring, lowering intracranial pressure, diverting ventricular blood, and allowing medication instillation [2,3,4]. If clinically indicated, an EVD collection system can also be accessed to withdraw cerebrospinal fluid for cultures or to obtain malignant cells. It is reported that over twenty thousand patients underwent EVD per annum in the United States, while the number may be even larger in countries such as China [3,4].

Intracranial infection is one of the most common and severe complications of EVD, with an incidence of 2–22% [5]. When EVD-associated intracranial infections are detected, a common consensus is to apply antibiotic therapy and/or replace the catheter with a new one, preferably at a new site [6,7]. Yet, even with dedicated therapy, EVD-associated intracranial infections may significantly affect patients’ outcomes by increasing both morbidity and mortality.

Great efforts have been made to screen features for predicting EVD-associated intracranial infections. It is hoped that identifying patients with high risks of infection may facilitate close surveillance and prophylactic treatment. Several features have been nominated by previous studies as potential risks for EVD-associated intracranial infections, including systemic infection, skull fracture, CSF leakage, lack of tunneling of a catheter, catheter irrigation, CSF sampling, the duration of EVD placement, and abnormalities of metabolism and nutrition states [8,9,10]. However, precise predictions are difficult for critical patients since the occurrence of EVD-associated intracranial infections may be influenced by multiple pathological processes, clinical procedures, and underlying diseases. It is also difficult for neurointensive care (NICU) specialists to manually screen every factor on top of a heavy workload. The application of machine learning in clinical research offers opportunities to build disease-prediction models and uncover hidden patterns from enormous datasets, which have been tested in many other clinical studies. Yet, how to select appropriate contributing features for model training and avoid the appearance of overfitting or underfitting are still issues that need to be addressed in various clinical studies. In this study, we assess the possibilities of developing prediction models for EVD-associated intracranial infections with the assistance of three popular machine-learning algorithms, the multivariate logistic regression, the support vector machine (SVM), and the K-nearest neighbor (KNN). Detailed methods and results are presented as follows. The least absolute shrinkage and selection operator (LASSO) regression was applied to help feature selection. Detailed methods and results are described as follows.

## 2. Methods

### 2.1. Study Cohort and Data Acquisition

This is a retrospective cohort study conducted between 1 January 2015 and 31 December 2020 at Shanghai Huashan hospital, Huashan Hongqiao Hospital, and Shanghai Jingan Hospital. The study received approval from the institutional ethics committee and was registered with the Chinese Clinical Trial Registry (ChiCTR1900021522). Due to the nature of a retrospective study, informed consent was waived. Data were collected from the hospital information system (HIS), electronic medical records (EMR), and laboratory information management system (LIS). Patients’ personal information was strictly protected by the ethics committee. The data set for analysis involves recordings of the demographic characteristics, diagnosis, medical history, laboratory tests on admission, and surgical procedures (if any).

### 2.2. Inclusion and Exclusion Criteria

The inclusion criteria involve patients who underwent EVD therapy with a diagnosis of traumatic brain injury (TBI), hydrocephalus, and nonaneurysmal spontaneous intracranial hemorrhage (ICH), which were confirmed by computed tomography (CT) or magnetic resonance (MR) scanning. Patients with confirmed preoperative intracranial infections were excluded. Patients with incomplete data or who died within 48 h after admission were also excluded. Figure 1 shows a flowchart of patient selection.

### 2.3. Definitions of EVD-Associated Infections and Length of EVD

The diagnosis of EVD-associated intracranial infections was established based on confirmed CSF-culture-positive bacterial meningitis or ventriculitis. The CSF for culture was collected through an EVD catheter or lumbar punctures. Once the diagnosis of EVD-associated intracranial infections is established, the catheter was removed as soon as possible. Depending on clinical need, the physician may decide whether to perform a lumbar puncture or reperform the EVD procedure. For patients without EVD-associated intracranial infections, physicians may decide when to remove the EVD catheter based on disease regression. The length of EVD was defined as the time between EVD catheter placement and removal. When there were multiple catheters (e.g., bilateral extraventricular drainage), the length of EVD was the time from first catheter placement to full catheter removal. For patients requiring catheter replacement due to EVD-associated intracranial infections, the length of EVD is counted from the time of initial catheterization to the replacement.

### 2.4. Statistics

Continuous variables were presented as mean ± standard deviation (SD) for normal distribution variables and median and interquartile range (IQR) for skewed distribution variables. For normality and homogeneity of variance, a Shapiro–Wilk W test and an F-test were conducted on the categorical variables. Student’s *t*-test, Wilcoxon rank-sum test, or one-way analysis of variance (ANOVA) were used for quantitative data from independent groups. To compare categorical variables, we used χ^2^ tests or Fisher exact tests. Statistical significance levels were all two-sided. Differences were considered statistically significant when *p* < 0.05. Statistical analyses were performed using R 3.6.3 (The R Foundation for Statistical Computing, Vienna, Austria) and Python 3.7 (The Python Software Foundation, Beaverton, OR, USA).

### 2.5. Feature Selection and Machine Learning Algorithms

Clinical features, including patients’ gender, age, medical history, diagnosis, American Society of Anesthesiologists grade (ASA), Glasgow Coma Scale (GCS) on admission, results of first hospitalization lab tests, the length of operation, and postoperative EVD monitoring variables, were collected. The ASA grades were determined preoperatively and collected from anesthesiology notes. A total number of 41 variables were analyzed as potential contributors to EVD-associated infection.

The multivariate logistic regression, SVM, and KNN algorithms were each applied to generate prediction models. Detailed descriptions of the three algorithms can be found in other studies and therefore are omitted in this paper. For multivariate logistic regression models, LASSO regression was applied to the training dataset for variable selection [11]. Candidate variables were excluded with a regression coefficient equal to zero after shrinkage. LASSO regression helps in feature selection. The LASSO model uses shrinkage and L1 regularization penalty technique, which means that the data points are recalibrated by adding a penalty to shrink the coefficients to zero if they are not substantial. For the SVM and KNN models, feature selection was conducted through weighting approaches using the Gini index and correlation calculation.

To prevent overfitting and maximize generalizability, we used fivefold cross-validation to train classifiers. The whole dataset was randomly divided into five roughly equally numbered subsets, each called a fold. Then, four of the five subsets were used as the training set and the remaining one as the validation set. The training used each of the five folds as the validation set, and the above process was repeated ten times.

The area under the receiver operating characteristic curve (AU-ROC), precision, classification accuracy, recall score, and F1 score were used to evaluate the performance of derived models. Accuracy = (TP + TN)/(TP + FP + FN + TN), recall = TP/(TP + FN), F1 = 2 × precision × recall/(precision + recall), T = true, F = false, P = positive, and N = negative. The ROC mean and SD of the model are calculated by repeated sampling five times.

### 2.6. Nomogram Construction

Features with statistical significance and served in the best performance model were applied to develop the nomogram. A nomogram is designed to predict the likelihood of an interested outcome and expresses the risk factors based on patients’ characteristics. A nomogram includes a points line for identifying the category of risk factors, a line of all risk factors indicating the points for each category, a total points line for the sum of each risk, and a probability line expressing the value of probability (usually from 0 to 100). The larger the points, the higher the likelihood of disease may occur. In the present study, the incidence of EVD-associated intracranial infections is a dependent variable, and the logistic regression model was used to construct the nomogram.

## 3. Results

### 3.1. Patients Characteristics

A total number of 3376 patients admitted to the NICU were retrospectively screened for analysis, among which 594 patients underwent EVD and with complete data were involved for analysis. Table 1 describes the demographics and clinical characteristics of recruited patients. Intracranial infections were confirmed in 143/594 (24.07%) patients with EVD. Patients with EVD-associated intracranial infections had longer hospital stays (24.49 ± 3.62 vs. 11.25 ± 3.75 days, *p* < 0.01), longer ICU length of stays (14.29 ± 6.91 vs. 5.93 ± 2.96 days, *p* < 0.01), and higher in-hospital mortality (32.87% vs. 5.76%). Table 1 summarized the difference between patients with or without EVD-associated intracranial infections. For further model training and derivation, the whole dataset was randomly divided into five subsets, four of which being used as the training set and the remaining one as the validation set. Appendix A indicated that there was no statistical difference between the training and the validation sets.

### 3.2. LASSO Regression and the Logistic Regression Model

From the potential EVD-associated intracranial infection risks listed in Table 1, candidate shrinkage was achieved by LASSO regression. Figure 2 indicates that during the LASSO regression, when the partial likelihood binomial deviance reached its minimum value, a value of 0. 038 was the optimal tuning parameter for LASSO regression. By this means, six features, namely HB, ASA grades, Lopt, Ab, a history of diabetes, and a diagnosis of traumatic SAH (tSAH) were selected as the best combination (Table 2). When these six features were introduced into the logistic-regression-model training, a prediction model was derived with an averaged AUC of 0.846 ± 0.006 (Table 3).

LASSO regression used least absolute shrinkage and selection operator (LASSO). Selection of tuning parameter (λ) in the LASSO regression used 5−fold cross-validation via minimum criteria (A-B). Partially unbiased binomial deviance was plotted vs. log (λ). At optimum log (λ), where features are selected, dotted vertical lines were drawn using the minimum criteria and the one standard error of the minimum criteria. A coefficient profile plot for each clinical feature was produced along with the log (λ) sequence. The dotted vertical line was set at nonzero coefficients, which were selected via 5−fold cross-validation, where six nonzero coefficients were included.

### 3.3. Models for EVD-Associated Infection Prediction

The feasibility of predicting EVD-associated intracranial infections was additionally evaluated using the SVM and KNN algorithms. According to the AUC values, logistic regression showed the best performance (AUC 0.846 ± 0.006) among the three tested approaches (Table 3, Figure 3A) in the training set. Table 3 shows that the logistic regression model also outperformed the other two algorithms in accuracy, sensitivity, specificity, positive predictive value, negative predictive value, and F1 score. The decision curve analysis (DCA) (Figure 3B) showed that when the risk threshold probability was set between 20% and 100%, the logistic regression model yielded the best prediction. The top six weighted features for prediction via the SVM and KNN models are listed in Table 2.

### 3.4. Model Validation

To assess the interpretability and generalizability of the prediction models, the three prediction models were further tested with the validation dataset. Table 4 and Figure 4 indicated that the logistic regression model still showed the best performance with an AUC of 0.847 ± 0.097 (Figure 4A). A linearized calibration curve showed that the logistic regression showed a nonstatistical difference with perfect predictions (on the 45° line, brief score = 0. 052, Figure 4B).

### 3.5. Nomogram Analysis for EVD-Associated Intracranial Infections Prediction

Figure 5 presents the nomogram constructed using the logistic regression model obtained from the results in Table 2. It is confirmed that the points of EVD-associated intracranial infections are high in the order of ASA grades, HB, history of diabetes, diagnosis of tSAH, length of operation, and Ab levels.

## 4. Discussion

For patients requiring neural intensive care, EVD is one of the most common life-saving procedures during the treatment of multiple brain injuries such as TBI, intracranial hemorrhage, and hydrocephalus [12]. The surgical procedure of EVD is relatively straightforward and can be performed by free hand. However, EVD is also one of the biggest concerns for NICU specialists due to its high incidence of intracranial infections [7,13]. Once occurred, EVD-associated intracranial infections are difficult to treat. Common treatments include catheter replacement, prolonged intravenous antibiotics administration, lumbar puncture, continuous CSF drainage, instill medications, and nutritional support. In certain circumstances, EVD-associated intracranial infections may develop into refractory drug-resistant bacterial meningitis, ventriculitis, systemic infections, and multiorgan failures that lead to unfavorable outcomes [14,15,16]. Though the exact figure is hard to calculate, it has been demonstrated that EVD-associated infection may significantly increase mortality, morbidity, and medical burden [17].

Considering its hazards and high incidence (2–22% according to the published literature) [5], scholars have tried to build predictive models for EVD-associated intracranial infections and find prevention strategies. Zhang et al. considered that ASA grades, length of hospital stay, consecutive operation, and prolonged surgery time may contribute to the occurrence of EVD-associated intracranial infections [18]. Yang and colleagues found that prolonged postoperative ICU stays, frequent CSF sampling, longer duration of EVD, and preoperative intubation were independent risk factors for EVD-associated intracranial infections [19]. However, these screened influences were either studied individually or reviewed as meta-analyses, which is not easy to be applied as systematic models.

In this study, we developed a prediction model for EVD-associated infection based on machine-learning algorithms. Established algorithms, including multivariable logistic regression, SVM, and KNN, were applied to generate prediction models. To avoid possible overfitting due to overloaded features, LASSO regression and five-fold cross-validation were adopted for variable selection and classification. The above three algorithms each have their own advantages and disadvantages. Logistic regression is a type of generalized linear model, which is suitable for fitting binary or multivalued data. In machine learning, logistic regression is considered a supervised learning approach, which can add or remove variables manually, and is not easily overfitted. The SVM algorithm can be applied to handle high-dimensional data sets, but it is sensitive to missing data. The KNN algorithm can be used for both classification and regression, but if the samples are not balanced, the bias of KNN prediction could be significant. In this study, the multivariable logistic model showed the highest AUC value, the best mean net benefit in DCA, and the smallest briefing score in the calibration curve. Based on the logistic regression model, HB, ASA grades, Lopt, Ab, a history of diabetes, and a diagnosis of tSAH were major contributors to predicting EVD-associated intracranial infections. To make the prediction model easier to assess and more practicable, a nomogram was established to give explicit weights of each selected feature [20,21].

The ASA classification is a widely used grading system for preoperative health of surgical patients. An ASA grade may range from 1 to 6, measuring a patient’s physical condition and risks of surgery. In this study, the ASA grades showed a significant difference between patients with and without EVD-associated intracranial infections and served as the top features for infection prediction in all three models. Patients with EVD-associated intracranial infections had an average higher ASA grade on admission, suggesting they might have more serious complications, more limited physical activity, a higher possibility of compromised immunity, and were more vulnerable to infections [22,23]. Though it is difficult to reverse unfavorable physical conditions immediately, efforts should be made to treat underlying diseases and complications, and to stabilize patients in critical conditions before an EVD or other operations.

Anemia is common among neurological critical patients. The decrease in HB level may be related to traumatic blood loss and/or complications of neurological disorders (e.g., gastrointestinal bleeding). A large amount of transfusion due to the correction of shock may also lead to a decrease in HB levels. Previous research has shown that decreased brain-tissue oxygen tension is an independent factor associated with unfavorable outcomes for TBI patients, while hemoglobin level is an indispensable factor for maintaining normal brain-tissue oxygen tension [24]. Oxygen is carried by the hemoglobin in the systemic circulation. Anemia reduces the oxygen-carrying capacity, which may lead to reduced oxygen uptake, prolonged edema, and elevated ICP after brain trauma or intracranial hemorrhage [25,26]. Those with hypohemoglobinemia may require extended ICU stay and drainage for ICP control, which increases the probability of EVD-associated intracranial infections.

Traumatic SAH is a common radiologic finding in CT scans, which occurred in 33–60% of TBI patients [27]. Causes of tSAH may involve the rupture of cortical vein and pia mater during brain contusion, with blood entering the subarachnoid space. Alternatively, cerebral arteries may also be injured, leading to hemorrhage due to sudden supination of the head during injury. While most patients with mild tSAH do not require surgical treatment, severe edema, vasospasm, and progressive lobar or intraventricular hemorrhage may develop in severe cases [28]. Hsieh reported that in contrast to patients with an isolated tSAH, those with tSAH and concurrent types of intracranial hemorrhage have higher mortality, and a large number of these cases require extended EVD for diverting intraventricular blood and continuous ICP monitoring [29]. In this study, it is indicated that the presence of tSAH suggests more severe brain injuries. The proportions of cases with GCS9–12 and GCS3–8 were significantly higher than those without tSAH, and patients with tSAH underwent longer EVD and had longer ICU/hospital stays (Appendix A). These could consequently contribute to the incidence of intracranial infections.

Serum albumin is used to predict the prognosis of TBI patients. In the general population, hypoalbuminemia has been associated with poor clinical outcomes in acute illnesses [30]. The level of serum albumin on admission has also been associated with in-hospital mortality, length of ICU stays, and readmission. The incidence of hypoalbuminemia among NICU patients is not rare, and early correction may help to reduce infection incidence and outcome improvement.

It is controversial whether the length of operation time may increase the chances of EVD-associated infections. While Yuen et al. reported no difference in length of operation time between infected and noninfected cases, others have found that longer operation time is associated with surgical-site infections and extracranial complications [31]. Although the procedure of EVD is usually straightforward, the operation time could be significantly extended when combining hematoma evacuation, decompression, and other procedures [32]. To some extent, the operation time may reflect the severity of brain injury, though whether the latter is associated with EVD-associated intracranial infections requires further investigations.

A history of diabetes serves as a high-weighted predictor for EVD-associated intracranial infections. Diabetes may cast negative influences on the rehabilitation progress of many diseases [33,34]. In patients with diabetes, the immune system may be compromised. In addition to causing natural barrier damage, diabetes can also impair cellular immunity due to insulin deficiency and hyperglycemia, causing the host to be more susceptible to infection [35,36].

In previous studies, clinical characteristics such as EVD duration were suggested as potential contributors for intracranial infections [8,9]. Our study showed that the average EVD duration for patients with intracranial infections was statistically longer than those without infections. The length of EVD was selected as a major predicting feature in the SVM model, but was excluded from the top features in the logistic regression and KNN models. One possible explanation is that the number of cases with extended periods of EVD in the current dataset was small. In our center, CSF drainage catheters were routinely removed once cerebral edema or SAH was relieved. For patients with hydrocephalus requiring emergency CSF drainage, subsequent shunts were arranged at the earliest opportunity. Therefore, although prolonged EVD has been indicated as a risk for intracranial infections, the importance was not identified within this dataset.

As with many other studies combining machine learning and disease models, we hope the findings of this study would be valuable for clinical practice and to improve patient outcomes. There are two key points in this study that make further translational research promising. First, the optimal algorithms and predictive models showed good and stable performance in both the training and validation dataset. Second, the six selected top-weighted predictors of the model are amenable to integration with clinical practice. For example, correcting preoperative hypoproteinemia and anemia, reducing EVD surgery time by standardized surgical training, and improving preoperative ASA scores by treating primary morbidity and complications may help to reduce the risk of EVD-associated intracranial infections. In addition, more attention should be paid to individuals with a history of diabetes mellitus and tSAH (or severe injuries), who are considered patients with high risks of EVD-associated intracranial infections. Further prospective clinical studies are needed to determine to what extent each of the above features is corrected (to derive cut-off values) and to what extent this correction can affect patients’ prognosis.

This research has some limitations. Firstly, no records were kept regarding patients’ courses after discharge and no information was available about patients’ outcomes when transferring hospitals. EVD-associated intracranial infections may be confirmed within rehabilitation hospitals after being discharged from NICU. Second, CSF-culture-negative infection may present under the effects of systemic antibiotics administration. The actual occurrence of EVD-associated infections could be higher than that reviewed in this manuscript. Third, despite the generation of an efficient prediction model, the prevention of EVD-associated infection is not easy. To reduce the occurrence of EVD-associated infection, a typical practice is to apply prophylactic antibiotic therapy covering typical skin flora during EVD. Yet, this may also contribute to the development of resistant organisms. Some other measures include reducing the frequency of CSF sampling, monitoring the EVD dressing site for possible leaks, maintaining the collection system upright, and not routinely replacing drain tubing. In some units, EVD catheters coated with antibiotic-impregnated and ionized silver particles have also been suggested to replace common catheters though they usually come at a cost. Yet, these measures, either alone or in combination, do not eliminate the occurrence of infection.

## 5. Conclusions

This study provides an example of a systematic analysis of data on EVD-associated intracranial infections in NICU patients. A practical prediction model has been generated using the logistic regression algorithm. ASA grade, serum Ab, duration of operation, diagnosis of tSAH, and a history of diabetes have been identified as major features for EVD-associated intracranial infections.

## Figures and Tables

**Figure 1 jcm-11-03973-f001:**
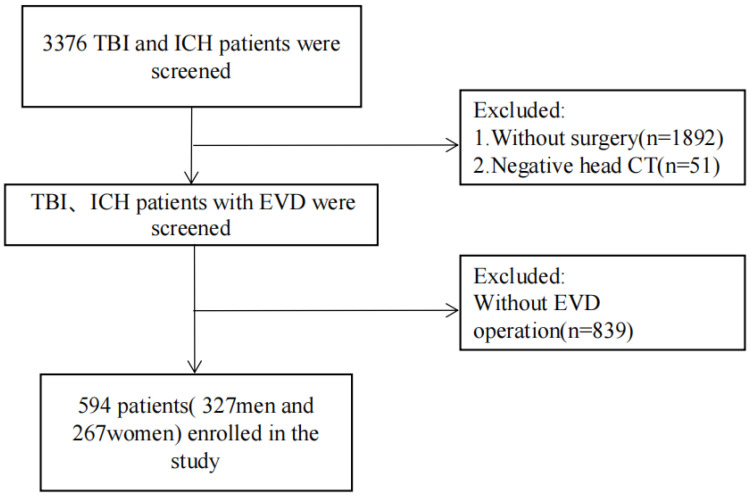
Flow chart for identifying eligible patients. Abbreviations: TBI, traumatic brain injury; ICH, intracranial hemorrhage; CT, computerized tomography; EVD, external ventricular drainage.

**Figure 2 jcm-11-03973-f002:**
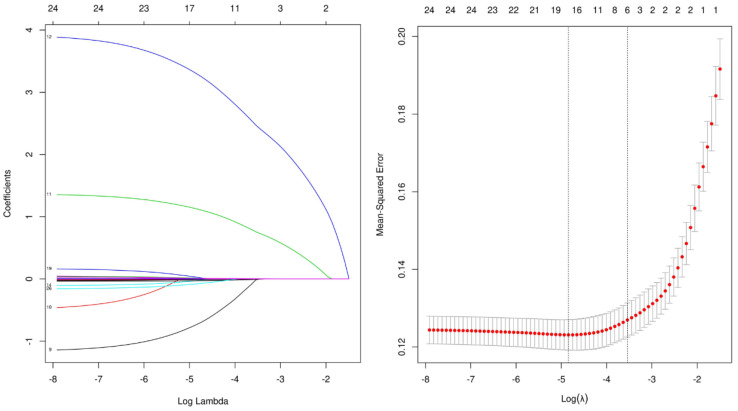
LASSO regression for feature selection. Different color lines represent different variables.

**Figure 3 jcm-11-03973-f003:**
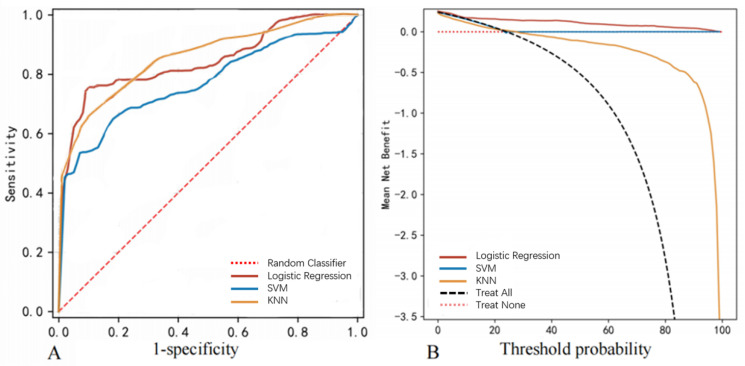
Model performance using the training dataset. (**A**) The ROC curve of the logistic regression, SVM, and KNN models. (**B**) The DCA of the logistic regression, SVM, and KNN models. The thick solid line signifies the assumption that no intracranial infection occurred in any patient on the *y*-axis.

**Figure 4 jcm-11-03973-f004:**
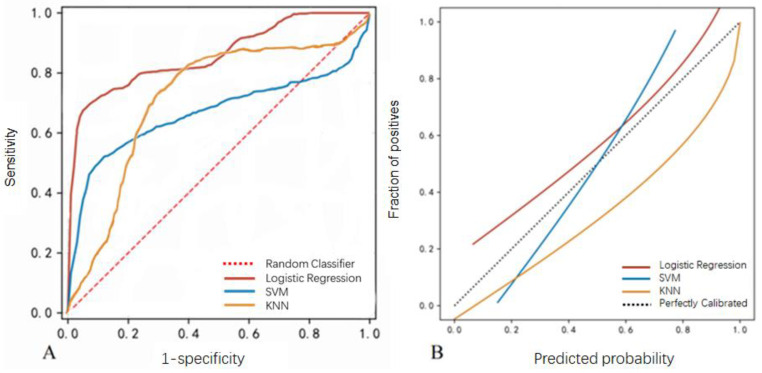
Model performance using the validation dataset. (**A**) The ROC curve of the logistic regression, SVM, and KNN models. (**B**) Calibration curves of the EVD-associated infection prediction. The *y*-axis meant the diagnosed infection. The *x*-axis meant the predicted risk of infection. The diagonal dotted line meant a perfect prediction by an ideal model. The solid line represented the performance of logistic regression, SVM, and KNN models, which indicated that a closer fit to the diagonal dotted line represented a better prediction.

**Figure 5 jcm-11-03973-f005:**
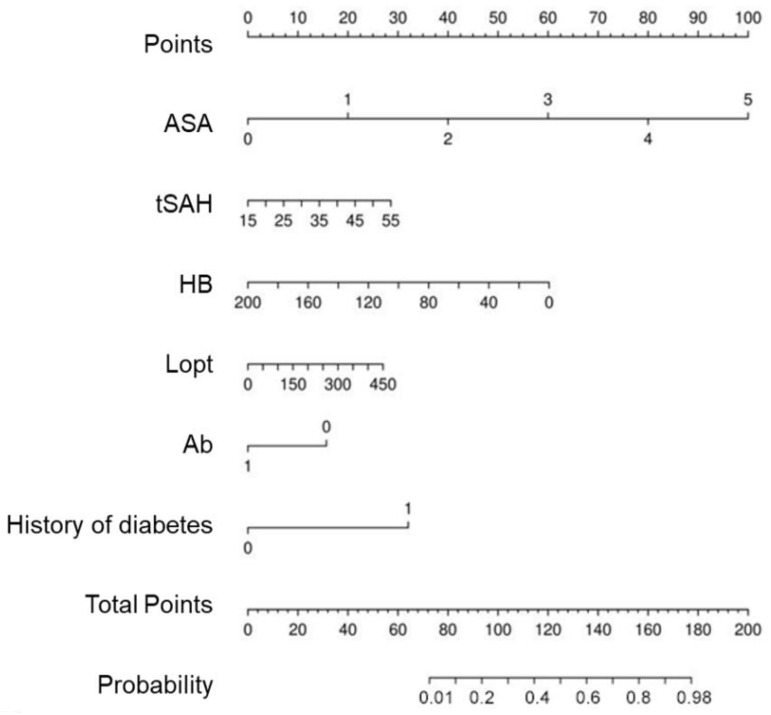
Development of the prediction nomogram. The EVD-associated infection risk nomogram was developed with the predictors including HB, ASA grades, Lopt, Ab, history of diabetes, and a diagnosis of tSAH.

**Table 1 jcm-11-03973-t001:** Demographic and clinical features of recruited patients *.

Variables	With EVD-Associated Intracranial Infections (*n* = 143)	Without EVD-Associated Intracranial Infections (*n* = 451)	*p*
Clinical characteristics			
Gender (Male)	77 (53.85%)	262 (58.09%)	0.36
Age (years)	54.08 (15.16)	55.28 (14.38)	0.39
BMI	23.22 (3.85)	23.48 (3.64)	0.48
History of diabetes	107 (74.83%)	337 (74.72%)	<0.01
ASA grades	3.04 (0.94)	2.25 (0.84)	<0.01
BP on admission (mmHg)	140.96 (12.30)	142.12 (12.55)	0.34
GCS 3–8 (*n*, %)	21 (14.69%)	70 (15.52%)	0.81
GCS 9–12 (*n*, %)	28 (19.58%)	87 (19.29%)	0.94
GCS 13–15 (*n*, %)	94 (65.73%)	294 (65.19%)	0.91
Preoperative intubation (*n*, %)	32 (22.38%)	84 (18.63%)	0.32
Lopt (minutes)	146.3 (92.44)	126.44 (79.85)	0.02
Cases underwent operations in addition to EVD (*n*, %)	62 (43.36%)	175 (38.80%)	0.33
Diagnosis and complications			
Hydrocephalus (*n*, %)	17 (11.89%)	56 (12.42%)	0.87
Spontaneous ICH (*n*, %)	95 (66.43%)	221 (49.00%)	<0.01
Traumatic brain injury (*n*, %)	62 (43.36%)	143 (31.71%)	0.01
Skull fracture (*n*, %)	41 (28.67%)	136 (30.16%)	<0.01
tSAH (*n*, %)	69 (48.25%)	146 (32.37%)	<0.01
CSF leakage due to trauma (*n*, %)	9 (6.29%)	27 (5.99%)	<0.01
Nonintracranial infections (*n*, %)	12 (8.39%)	28 (6.21%)	0.36
First laboratory tests			
RBC (10^12^/L)	4.49 (0.89)	4.50 (0.83)	0.88
HB (g/L)	75.28 (61.83)	117.85 (32.26)	<0.01
WBC (10^9^/L)	13.64 (4.24)	12.88 (5.15)	0.08
NEUT (%)	86.01 (6.21)	83.83 (10.58)	<0.01
PLT (10^9^/L)	200.86 (65.21)	202.49 (68.66)	0.8
TBIL (μmol/L)	12.28 (6.92)	11.39 (6.84)	0.18
DBIL(μml/L)	5.15 (2.91)	4.94 (2.83)	0.44
ALT (U/L)	43.17 (38.24)	33.12 (23.60)	<0.01
AST (U/L)	43.23 (27.85)	38.32 (29.57)	0.15
LDH (U/L)	206.94 (69.11)	205.53 (64.56)	0.82
HDL (mmol/L)	1.93 (0.59)	1.96 (0.62)	0.62
LDL (mmol/L)	2.93 (1.15)	2.97 (1.15)	0.77
Ch (μml/L)	5.07 (1.18)	5.03 (1.15)	0.75
Ab (g/L)	33.27 (7.22)	43.74 (4.62)	<0.01
GLB (g/L)	30.71 (6.62)	32.02 (7.61)	0.01
BUN (mmol/L)	6.09 (1.77)	6.05 (1.76)	0.79
UA (mmol/L)	261.83 (101.63)	250.59 (94.69)	0.23
SCR (μmol/L)	5.87 (1.21)	6.03 (1.18)	0.16
Postoperative EVD monitoring			
Length of EVD (days)	9.15 (4.52)	6.85 (3.48)	0.045
Number of CSF sampling (per week)	3.07 (1.41)	3.00 (1.41)	0.62
Leakage from EVD site (*n*, %)	20 (13.99%)	23 (5.10%)	<0.01
Outcomes			
ICU length of stays (days)	14.29 (6.91)	5.93 (2.96)	<0.01
Hospital stays (days)	24.49 (3.62)	11.25 (3.75)	<0.01
In-hospital mortality (*n*, %)	47 (32.87%)	26 (5.76%)	<0.01

* Continuous data are shown as mean (standard deviation). Abbreviations: EVD, external ventricular drainage; CSF, cerebropinal fluid; ICH, intracranial hemorrhage; ASA, American Society of Anesthesiologists; Lopt, length of operation time; tSAH, traumatic subarachnoid hemorrhage; RBC, red blood cell; HB, hemoglobin; WBC, white blood cell; NEUT, neutrophil ration; PLT, platelet; TBIL, indirect bilirubin; DBIL, direct bilirubin; ALT, glutamic pyruvic transaminase; AST, glutamic oxalacetic transaminase; LDH, lactate dehydrogenase; HDL, high-density lipoprotein; LDL, low-density lipoprotein; Ch, cholinesterase; Ab, albumin; GLB, globulin; BUN, urea nitrogen; UA, uric acid; SCR, creatinine; BP, blood pressure; GCS, Glasgow coma scale; BMI, body mass index; ICU, intensive care unit.

**Table 2 jcm-11-03973-t002:** Top weighted features for predicting EVD-associated intracranial infections.

Models	Features	Weight
Logistic Regression	ASA grades	0.37
	HB	0.24
	History of diabetes	0.19
	tSAH	0.092
	Lopt	0.087
	Ab	0.068
SVM	HB	0.31
	ASA grades	0.29
	Ab	0.18
	Lopt	0.16
	History of diabetes	0.13
	Length of EVD	0.076
KNN	ASA grades	0.37
	HB	0.17
	ALT	0.16
	Lopt	0.14
	Ab	0.095
	Leakage from EVD site	0.075

Abbreviations: ASA, American Society of Anesthesiologists; SVM, Support Vector Machine; KNN, K-nearest neighbor; EVD, external ventricular drainage; Lopt, length of operation time; HB, hemoglobin; tSAH, traumatic subarachnoid hemorrhage; ALT, glutamic pyruvic transaminase; Ab, albumin.

**Table 3 jcm-11-03973-t003:** Multimodel classification—training cohort *.

Classification Model	AUC	Cut-Off	Accuracy	Sensitivity	Specificity	Positive Predictive Value	Negative Predictive Value	F1 Score
Logistic regression	0.846 (0.006)	0.305 (0.016)	0.870 (0.004)	0.761 (0.010)	0.970 (0.005)	0.715 (0.011)	0.921 (0.003)	0.737 (0.008)
SVM	0.730 (0.008)	0.191 (0.100)	0.792 (0.029)	0.646 (0.059)	0.845 (0.054)	0.575 (0.097)	0.879 (0.009)	0.599 (0.018)
KNN	0.845 (0.003)	0.400 (0.001)	0.887 (0.005)	0.931 (0.014)	0.817 (0.012)	0.828 (0.021)	0.901 (0.007)	0.876 (0.012)

* All values are shown as mean (standard deviation). Abbreviations: AUC, Aera Under Curve; SVM, support vector machine; KNN, k-nearest neighbor.

**Table 4 jcm-11-03973-t004:** Multimodel classification—validation cohort *.

Classification Model	AUC	Cut-Off	Accuracy	Sensitivity	Specificity	Positive Predictive Value	Negative Predictive Value	F1 Score
Logistic regression	0.847 (0.097)	0.305 (0.016)	0.869 (0.050)	0.787 (0.145)	0.923 (0.058)	0.714 (0.094)	0.924 (0.048)	0.743 (0.101)
SVM	0.677 (0.111)	0.191 (0.100)	0.758 (0.069)	0.698 (0.174)	0.875 (0.135)	0.521 (0.140)	0.856 (0.050)	0.585 (0.122)
KNN	0.844 (0.072)	0.400 (0.000)	0.829 (0.053)	0.833 (0.136)	0.747 (0.156)	0.730 (0.170)	0.859 (0.051)	0.757 (0.100)

* All values are shown as mean (standard deviation). Abbreviations: AUC, Aera Under Curve; SVM, support vector machine; KNN, k-nearest neighbor.

## Data Availability

The data presented in this study are available on request from the corresponding author. The data are not publicly available due to patient privacy protection.

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
