# Peer review of "Prediction of Intracranial Infection in Patients under External Ventricular Drainage and Neurological Intensive Care: A Multicenter Retrospective Cohort Study"

_jcm, 2022, doi:10.3390/jcm11143973_

Round 1

Reviewer 1 Report

Fu and colleagues address the issue of EVD infection using advanced statistical methods.  The authors perform a retrospective investigation of variables that may contribute to infection, and utilize classical logistic regression analysis, as well as machine learning algorithms SVM and KNN. 

1.       The question arises why duration of the EVD does not seem to have an impact on the likelihood of infection, as substantial literature exists on this topic.  In Table 2, how was the variable “length of EVD” measured? (i.e. in cases of infection, was this length of time until the infection; in cases of no infection, was this total EVD duration?).  If data was expressed differently (for instance, comparing all EVDs at day 7 compared to day 14 for presence of infection), one could imagine that the overall prevalence of infection at day 14 would be higher than at day 7—however, in the manner that the data is currently expressed (comparison of average EVD duration), it’s possible that a key variable is being suboptimally evaluated. 

2.       Text is lacking in the discussion regarding the extent to which infection can be predicted by the authors’ approach, and the expected clinical utility.  Understanding that the role of predictive data would lie in patient counseling (as the authors suggest), would a positive predictive value of 0.714-0.730 be expected to change clinical practice? 

Additionally, concern arises with regard to the utility of machine learning, given that the standard statistical approach (multivariate logistic regression) seems to outperform the SVM and KNN models in certain respects, although this is not directly addressed in the discussion.  Why would a clinician benefit from using results from machine learning if logistic regression is satisfactory?

3.       Selection of variables: In the discussion section, paragraphs 1 and 2, a number of variables determined to be historically relevant were alluded to, including CSF sampling rate, ICU length of stay, hospital stay, consecutive operation, pre-operative intubation, presence/absence of CSF leak, possibility of leaking from EVD site, presence/absence of hydrocephalus (as a contributor to leaking when EVD clamped).  It does not appear that any of these variables were included in the analysis (Tables 1 and 2).  Were these excluded for a reason?

Also, why isn’t mechanism of injury (aneurysmal rupture, spontaneous ICH, trauma, hydrocephalus, tumor) included among the variables?

4.       Throughout the text, reference is made to the importance of ASA grade in the predictive set of variables, however ASA grade does not appear to be in the variables listed in Table 1 or Table 2.  This should be addressed.

Additionally with regard to ASA grade, the authors’ nomogram (Figure 5) places a substantial amount of weight on the ASA grade (a difference between ASA 3 and ASA 4 confers 20 points).  One could forsee that ASA grade might be subjective in the neurosurgical population (grade 3 vs 4: severe systemic illness vs. severe systemic illness threatening to life), and as many/most neurosurgical conditions may be considered variably life-threatening, and the methods of ASA grade determination become quite relevant, with large impact on the results of the model.  Were these ASA grades determined pre-operatively? Post-operatively? This makes a difference because the ASA grade prior to surgical decompression of a mass lesion would be different from the ASA grade assessed after surgical stabilization and during the process of EVD weaning.  Are these values recorded from anesthesiology notes or determined subjectively by the research group and at what time during the clinical course?  This would seem to make an impact. 

Specific Comments:

11.       Authors discuss tSAH (lines 281-288) as a contributing variable, although it may not be immediately obvious to the reader why tSAH presence impacts infection rate.  Clinical correlation should be included as to why tSAH would contribute to infection (are we to assume that it is a marker of injury severity?). Would like to ensure that the authors are not combining traumatic SAH with arterial/aneurysmal SAH (which would a separate phenomenon, correlated with a presumably different EVD duration and level of morbidity). 

22.       The display format in Table 2 appears to alternate between using ( ) to indicate error range (SD), and also using ± to list error.

33.       The numerical calculations in Table 1 are potentially inaccurate.  In the last row, EVD-associated infection is 51 (9.7%).  However, 51/594=8.6%, not 9.7%.  May need to check all other values for erroneous calculation.

44.       In table 2: is the rate of infection in the training set n=119/483=25%, and the rate of infection in the validation set 21%?  Unclear if this is simply confusion regarding the pruning of the dataset, because this implies a much higher infection rate 25% and 21% compared to the overall total from Table 1 (8-9%).  

55.       In Table 2, validation set: tSAH 28 has no percentage.  Was this intentional?

Author Response

Dear editor,

Response to Reviewers (JCM-1756485)

Thank you and the reviewer’s suggestions on our manuscript. We have carefully studied the reviewers’ suggestions and revised our manuscript. A point-to-point reply is provided as follows.

  1. The question arises why duration of the EVD does not seem to have an impact on the likelihood of infection, as substantial literature exists on this topic.  In Table 2, how was the variable “length of EVD” measured? (i.e. in cases of infection, was this length of time until the infection; in cases of no infection, was this total EVD duration?). … it’s possible that a key variable is being suboptimally evaluated.

Response:

In this study, the length of EVD was defined as the time between EVD catheter placement and removal. When there were multiple catheters (e.g., bilateral EVD), the length of EVD was the time from first catheter placement to full catheter removal. Our data showed that the average EVD duration for patients with intracranial infections was statistically longer than those without infection. The length of EVD was selected as a major predicting feature in the SVM model, but was excluded from the top features in the logistic regression and KNN model. One possible explanation is that the number of cases with extended periods of EVD was small. In our center, CSF drainage catheters were routinely removed once cerebral edema or SAH is relieved. For patients with hydrocephalus requiring emergency CSF drainage, subsequent shunts were arranged at the earliest opportunity. Therefore, although prolonged EVD has been indicated as an important risk for intracranial infections, the importance was not identified within this dataset. The definition of EVD duration has now been clarified in the Methods section. The finding of EVD duration on intracranial infections have been supplemented in Table 1 and Table 2. We also discussed the reasons why it is not selected as a major contributor to EVD-associated infections in the Discussion section.  

  1. Text is lacking in the discussion regarding the extent to which infection can be predicted by the authors’ approach, and the expected clinical utility.

Response:

 (1) In this study, we only investigated the incidence of EVD-associated intracranial infections and their related risks. Therefore, our results may help to predict the incidence of EVD-associated intracranial infections. In the previous version of the manuscript, there were several places where we did not specify intracranial infections, which may have caused some misunderstanding. These have now all been corrected. Yet, due to the limitation of the retrospective study and data collection, we could not identify pathological subtypes (e.g. encephalopyosis, ventriculitis, meningitis) or pathogens of each intracranial infection case, which was worthy of further study in the future.

 (2) We hope the findings of this study would be useful for clinical practice. First, the optimal prediction model (logistic regression model) demonstrates a well predictive power with an AUC value of 0.846±0.006. Second, among the six screened major risk factors, some physiological indicators (hemoglobin and albumin levels) could be adjusted before EVD or during the perioperative period at the earliest opportunity. It may also be possible to shorten the length of operations through proper training or optimizing the process, which may help to reduce the incidence of EVD-related intracranial infections. In addition, extra attention may be required for patients with a high ASA grade, diagnosis of tSAH, and a history of diabetes mellitus. Based on the results of this study, we are also interested in deriving cut-off values for each weighted feature or building a scale system, though this may require future studies with a larger sample size.

We have added the above two points into the Discussion part.   

  1. Additionally, concern arises with regard to the utility of machine learning, given that the standard statistical approach (multivariate logistic regression) seems to outperform the SVM and KNN models in certain respects. Why would a clinician benefit from using results from machine learning if logistic regression is satisfactory?

Response:

Logistic regression is a type of generalized linear model, which is suitable for fitting binary or multi-valued data. In machine learning, logistic regression is considered a supervised learning approach and is often used as a classification model. Therefore, logistic regression is not exclusive to other algorithms such as SVM and KNN. Logistic regression is easy to implement, can add or remove variables manually, and is not easily overfitted. The SVM algorithm can be applied to handle high-dimensional data sets, but it is sensitive to missing data. The KNN algorithm can be used for both classification and regression, but if the samples are not balanced, the bias of KNN prediction could be significant. In this study, the logistic model has the highest AUC value, the best Mean Net Benefit in DCA, and the smallest briefing score in the calibration curve. therefore it performs best among the three algorithms. For the selection of machine learning algorithms, we have made additional discussion in the Discussion section.

  1. Selection of variables: In the discussion section, paragraphs 1 and 2, a number of variables determined to be historically relevant were alluded to, including CSF sampling rate, ICU length of stay, hospital stay, consecutive operation, pre-operative intubation, presence/absence of CSF leak, possibility of leaking from EVD site, presence/absence of hydrocephalus (as a contributor to leaking when EVD clamped).  It does not appear that any of these variables were included in the analysis (Tables 1 and 2).  Were these excluded for a reason?

Response:

Clinical information including CSF sampling rate, ICU length of stay, hospital stay, consecutive operation, pre-operative intubation, CSF leakage, and presence of hydrocephalus have now been collected (Table 1) and involved in the analysis. We have supplemented this information in Table 1 and Table 2.

  1. Also, why isn’t mechanism of injury (aneurysmal rupture, spontaneous ICH, trauma, hydrocephalus, tumor) included among the variables?

Response:

Mechanisms of injury including spontaneous ICH, trauma, and hydrocephalus have been included for analysis (Table 1 and Table 2). Yet, since this study only involved patients with traumatic brain injury, hydrocephalus, and non-aneurysmal spontaneous intracranial hemorrhage, aneurysmal rupture and tumor were not involved in this study.

  1. Throughout the text, reference is made to the importance of ASA grade in the predictive set of variables, however ASA grade does not appear to be in the variables listed in Table 1 or Table 2.  This should be addressed...Were these ASA grades determined pre-operatively? Post-operatively?This makes a difference because the ASA grade prior to surgical decompression of a mass lesion would be different from the ASA grade assessed after surgical stabilization and during the process of EVD weaning.  Are these values recorded from anesthesiology notes or determined subjectively by the research group and at what time during the clinical course?

Response:

The data of ASA has now been supplemented in both Table 1 and Table 2. The ASA grades showed a significant difference between patients with and without EVD-associated intracranial infections and served as the top features for infection prediction in all three models. The ASA grades were determined pre-operatively and collected from anesthesiology notes. This information has now been added into the Methods section.

  1. Authors discuss tSAH (lines 281-288) as a contributing variable, although it may not be immediately obvious to the reader why tSAH presence impacts infection rate.  Clinical correlation should be included as to why tSAH would contribute to infection (are we to assume that it is a marker of injury severity?). Would like to ensure that the authors are not combining traumatic SAH with arterial/aneurysmal SAH (which would a separate phenomenon, correlated with a presumably different EVD duration and level of morbidity). 

Response:

In this study, patients with intracranial hemorrhage due to aneurysm rupture were not recruited. Therefore tSAH in this study means traumatic SAH only. To better illustrate the relationship between tSAH, injury severity, and its contributions to EVD-associated intracranial infections, a Supplementary Table 2 has been provided. It is indicated that the presence of tSAH suggests more severe injury. The proportions of cases with GCS9-12 and GCS3-8 were significantly higher than those without tSAH, and patients with tSAH underwent longer EVD and had longer ICU/hospital stays (Supplementary Table 2), which could consequently contribute to the incidence of intracranial infections.  

  1. The display format in Table 2 appears to alternate between using ( ) to indicate error range (SD), and also using ± to list error.

Response:

The inconsistent display format has now been corrected throughout the manuscript. The SD values are displayed as (SD) in tables. 

  1. The numerical calculations in Table 1 are potentially inaccurate.  In the last row, EVD-associated infection is 51 (9.7%).  However, 51/594=8.6%, not 9.7%.  May need to check all other values for erroneous calculation.

Response:

We have double-checked the results and corrected previous typos and miscalculations. Tables have been updated in the manuscript. 

  1. In table 2: is the rate of infection in the training set n=119/483=25%, and the rate of infection in the validation set 21%?  Unclear if this is simply confusion regarding the pruning of the dataset, because this implies a much higher infection rate 25% and 21% compared to the overall total from Table 1 (8-9%). In Table 2, validation set: tSAH 28 has no percentage.  Was this intentional?

Response:

We apologized for the previous incorrect presentation of the data. The data has been updated and corrected in the manusctript. A new supplementary Table 1 has been provided to illustrate the difference between the training and validation datasets.

We appreciate the great efforts and the valuable suggestions from both the editor and the reviewers. We are now ready to submit a revised manuscript and hope this version would be suitable for publication. If further revisions are required, please let us know.

Best wishes

Yirui Sun M.D., Ph.D.

For all authors.

Reviewer 2 Report

The authors provide an innovative manuscript regarding EVD-associated infections; the use of machine-learning is interestingly applied to elaborate this model. 

My principal criticisms to this work are related to translational relevance: how do the authors think that this study can reduce EVD-related infections? How do they purpose a correct management of this situation (I.e. correction of unfavorable factors or early substitution of EVD). Furthermore, how about future perspectives? How does this research can become useful for intensive patient management?

Author Response

Dear editor,

Response to Reviewers (JCM-1756485)

Thank you and the reviewer’s suggestions on our manuscript. We have carefully studied the reviewers’ suggestions and revised our manuscript. A point-to-point reply is provided as follows.

My principal criticisms to this work are related to translational relevance: how do the authors think that this study can reduce EVD-related infections? How do they purpose a correct management of this situation (I.e. correction of unfavorable factors or early substitution of EVD). Furthermore, how about future perspectives? How does this research can become useful for intensive patient management? 

Response:

We apricate the concerns on the translational application, which is similar to question 2 from Reviewer 1. As many other studies combing machine learning and disease models, we hope the findings of this study would be valuable for the clinical practice and to improve patient outcomes. There are two key points in this study that make further translational research promising. First, the optimal algorithms and predictive models showed good and stable performance in both the training and validation dataset. Second, the six selected top-weighted predictors of the model are amenable to integration with clinical practice. For example, correcting pre-operative hypoproteinemia and anemia, reducing EVD surgery time by standardized surgical training; and improving preoperative ASA scores by treating primary morbidity and complications may help to reduce the risk of EVD-associated intracranial infections. In addition, more attention should be paid to individuals with a history of diabetes mellitus and tSAH (or severe injuries), who are considered patients with high risks of EVD-associated intracranial infections. Further prospective clinical studies are needed to determine to what extent each of the above features is corrected (to derive cut-off values) and to what ex-tent this correction can affect patients' prognosis. Additional discussion on the value of the application of this study is provided in the Discussion section.

We appreciate the great efforts and the valuable suggestions from both the editor and the reviewers. We are now ready to submit a revised manuscript and hope this version would be suitable for publication. If further revisions are required, please let us know.

Best wishes

Yirui Sun M.D., Ph.D.

For all authors.

Round 2

Reviewer 1 Report

Acceptable in current form